# Genus *Nocardiopsis*: A Prolific Producer of Natural Products

**DOI:** 10.3390/md20060374

**Published:** 2022-05-31

**Authors:** Ting Shi, Yi-Fei Wang, Han Wang, Bo Wang

**Affiliations:** 1College of Chemical and Biological Engineering, Shandong University of Science and Technology, Qingdao 266590, China; shiting_jia@126.com (T.S.); kingsley11f115@163.com (Y.-F.W.); h15725209196@163.com (H.W.); 2State Key Laboratory of Microbial Technology, Institute of Microbial Technology, Shandong University, Qingdao 266200, China

**Keywords:** actinomycetes, *Nocardiopsis*, natural products, bioactivities, medicinal potentiality

## Abstract

Actinomycetes are currently one of the major sources of bioactive secondary metabolites used for medicine development. Accumulating evidence has shown that *Nocardiopsis*, a key class of actinomycetes, has the ability to produce novel bioactive natural products. This review covers the sources, distribution, bioactivities, biosynthesis, and structural characteristics of compounds isolated from *Nocardiopsis* in the period between March 2018 and 2021. Our results reveal that 67% of *Nocardiopsis*-derived natural products are reported for the first time, and 73% of them are isolated from marine *Nocardiopsis*. The chemical structures of the *Nocardiopsis*-derived compounds have diverse skeletons, concentrating on the categories of polyketides, peptides, terphenyls, and alkaloids. Almost 50% of the natural products isolated from *Nocardiopsis* have been discovered to display various bioactivities. These results fully demonstrate the great potential of the genus *Nocardiopsis* to produce novel bioactive secondary metabolites that may serve as a structural foundation for the development of novel drugs.

## 1. Introduction

Actinomycetes belong to Gram-positive bacteria and are one of the biggest bacterial phyla [1,2]. The high G+C DNA content of actinomycetes implied their enormous biosynthetic potential to produce various natural products with diverse structures and important commercial applications [1,2]. Two-thirds of all naturally derived antibiotics have been discovered from actinobacteria [3,4]. Approximately 70% of the pharmaceutically active natural products which are currently used in clinics are isolated from actinobacteria [5,6,7,8], including a series of anticancer, antifungal, antibacterial, antihelminthic, and immunosuppressive drugs [3].

*Nocardiopsis* is an important genus of actinobacterium for its extensive application in agriculture [9], industry [10], and environmental protection [11], especially for its potential ability to produce new natural products [5,12]. By mid-2021, 3% of marine actinomycetes-derived natural products were produced by *Nocardiopsis*, and *Nocardiopsis* are the third-largest actinomycetes in terms of producing marine compounds [13]. In addition, *Nocardiopsis* are a prolific source of bioactive natural products in both marine and terrestrial environments [1] and are widely distributed in multiple ecosystems [12], including deserts [14], deep ocean [15], coastal wetlands [16], and saline–alkali soil [17]. *Nocardiopsis* species have the ability to survive under different and hostile environmental conditions, mainly benefitting from their excretion of enzymes, multipurpose genetic constitution, and their ability to produce compuponible solutes and surfactants [12,18,19,20]. Besides these characteristics, the members of this genus have the ability to produce abundant bioactive compounds, which may allow them to prevail in different habitats [12]. Extracts from cultivated *Nocardiopsis* have exhibited cytotoxic [21,22], antimicrobial [23,24], antifibrotic [25], and anti-inflammatory [25] activities. Comprehensive analyses of *Nocardiopsis* metabolites by LC-HRES-MS, HRMS, or GC/MS have led to the identification of bioactive and structurally diverse compounds [21,26,27,28]. The bioactive secondary metabolites isolated from *Nocardiopsis* include antimicrobial agents [29,30,31], tumor promoters [32], cytotoxic compounds [33,34], kinase and P-glycoprotein inhibitors [35,36], immunoregulators [37], and natural products with other multiple bioactivities [38,39,40]. The secondary metabolites discovered from *Nocardiopsis* have shown a great diversity of structural frameworks, including polyketides [41,42,43], alkaloids [44,45], and terpenoids [46,47].

Bennur et al. reviewed the bioactive natural products derived from *Nocardiopsis* prior to 2015 [12], while Ibrahim et al. conducted a literature review regarding all of the secondary metabolites discovered from *Nocardiopsis* from 2016 to February 2018 [5]. In this review, the sources, distribution, bioactivities, biosynthesis, and structural characteristics of the compounds isolated from *Nocardiopsis* between March 2018 and 2021 are comprehensively summarized.

## 2. Polyketides

Three new angucyclines, nocardiopsistins A–C (**1**–**3**), were obtained from the deep-sea sponge-derived *Nocardiopsis* sp. strain HB-J378 (Figure 1) [48]. The antibacterial activities of compounds **1**–**3** were tested against MRSA (methicillin-resistant *Staphylococcus aureus*) and **1**–**3** exhibited antibacterial activity with MIC values of 3.12–12.5 μg/mL. Three core genes were identified through bioinformatic analysis of the sketch genome of the strain HB-J378 in a biosynthetic gene cluster encoding a typical aromatic or type II polyketide synthase (PKS) system, including acyl carrier protein (ACP), ketoacyl: ACP synthase *α*-subunit (KS*_α_*) and *β*-subunit (KS*_β_*). The brief biosynthetic route for **1**–**3** was proposed according to the discovered oviedomycin pathway [49]. Compounds **1**–**3** were supposed to be biosynthesized by taking advantage of one molecule of isobutyral-CoA and nine molecules of malonyl-CoA, through complex enzymatic reactions to obtain the key angucycline biosynthetic intermediate UWM6, an analog of **1**–**3** [50], to acquire **1**–**3** (see Figure 1) [48].

The new strain *Nocardiopsis* sp. CG3 (DSM 106572), collected from the saltpan of Kenadsa (22 km west of Bechar, located in southwest Algeria), was discovered to have the ability to produce new bioactive natural products in a screening program [51]. Chemical investigation of the strain led to the isolation of five new polyene macrolactams, kenalactams A−E (**4**−**8**) (Figure 2) [51]. The biosynthetic pathway of polyketide kenalactam A (**4**) was studied by feeding experiments and was found to used L-alanine as the nitrogen-bearing starter unit. Compounds **6**−**8** exhibited cytotoxic activity against HeLa (cervical cancer cells KB3.1) and PC-3 (human prostate cancer) cell lines, with IC_50_ values ranging from 2.1 to 6.8 μM [51].

Eight new *α*-pyrone derivatives, nocahypyrones A–H (**9**–**16**), along with one known analog, germicidin G (**17**) (Figure 3) were isolated from the strains *Nocardiopsis* sp. HDN154-146 and HDN154-168, which were collected from soil samples derived from the Takla Makan desert area in Xinjiang Province, China. This was the first time it was reported that compounds **13** and **16** showed cytoprotective activity by inducing the expression of phase II detoxifying enzymes [52]. Aldo-keto reductase family1 member C1 (AKR1C1), human NAD(P)H: quinone oxidoreductase 1 (NQO1), superoxide dismutase 2 (SOD2), and heme oxygenase 1(HO-1), belonging to phase II detoxifying enzymes, have been illustrated to possess significant roles in defending mammalian cells against oxidative damage and excessive inflammatory reaction [53,54]. Compounds **13** and **16** displayed cytoprotective activity by inducing the expression level of SOD2 and HO-1 in HaCaT cells [52].

Chemical investigation of the actinobacterium *Nocardiopsis* sp. HDN 17-237 led to the isolation of one new *β*,*γ*-butenoate derivative, phenylbutenote (**18**), and one new *α*-pyrone, nocapyrone T (**19**) (Figure 3). The strain HDN 17-237 was collected from deep-sea water from the Mariana Trench (depth 4448 m, 10°21.100′ N, 142°17.574′ E, gathered in September 2016). Compounds **18** and **19** were evaluated for antibacterial and antioxidant activities, while neither of them exhibited obvious activity [55].

Comprehensive research of the secondary metabolites of an Antarctic marine animal sample-derived actinomycete, *N. aegyptia* HDN19-252, combined with a molecular networking approach in the Global Natural Products Social (GNPS) platform, obtained the isolation of four new anthraquinone derivatives, saliniquinones G−I (**20**–**22**) and heraclemycin E (**23**) (Figure 3). Compounds **20** and **21** showed potent antibacterial activities against six evaluated bacterial strains—*Bacillus subtilis*, methicillin-resistant coagulase-negative *staphylococci* (MRCNS), *B. cereus*, *Proteus* sp., *Mycobacterium phlei*, and *Escherichia coli*—with MIC values of 3.1−12.5 µM [56].

Three new macrolides, borrelidins C−E, along with one known analog, borrelidin, which showed antibacterial and anticancer activities, were isolated from a saltern-derived halophilic *Nocardiopsis* sp. in 2017 [43]. These borrelidins were found to have the ability to relieve the amyloid-*β* induced toxicity in the HT22 cell line in 2021, indicating the potential of borrelidins to be developed as Alzheimer’s disease drugs [57].

## 3. Alkaloids

The approach of an optimized nitroso-based probe (dienophile probe), promoting the detection of compounds containing conjugated alkenes in crude broth extracts, was employed in the chemical investigation of a marine-derived *Nocardiopsis* sp. CNY-503 and gained the isolation of one new polyketide alkaloid, named nocarditriene (**24**), which contained an unprecedented epoxy-2,3,4,5-tetrahydropyridine structure (Figure 4) [58].

The structure of nocarbenzoxazole G (**25**), isolated from the marine-derived actinomycete *N. lucentensis* DSM 44048 in 2015 [60], was revised into the nocarbenzoxazole G (**26**) molecule by total synthesis in 2019. The benzoxazole skeleton was constructed with microwave assistance and continued by carbon–carbon bond formation with relevant aryl bromides [59]. Compound **26** was found to display moderate cytotoxicity against HepG2 and HeLa cell lines with IC_50_ values of 16 and 14 μM, respectively [60].

Chemical investigation of the strain *N. flavescens* NA01583, which was gained from marine sediment gathered at the coast near Hainan Island in 2016 through the genome mining of an indolocarbazole-type gene cluster, led to the isolation of three new indolocarbazole alkaloids, named loonamycins A−C (**27**−**29**) (Figure 5) [61].

Compound **29** was produced successfully by the allogenetic expression of the complete *loo* gene cluster in a vicarious host, *Streptomyces lividans* K4−114. The indolocarbazole skeleton of **27**−**29**, belonging to the family of indolocarbazole alkaloids, is structurally similar to that of rebeccamycin and staurosporine with an additional rare modified tryptophan ring. The molecular bases of these modifications were investigated by carefully analyzing *loo* BGC for further genome mining and combinational biosynthetic research. The *loo* gene cluster sustains a ∼36 kb continuous DNA sequence, including 25 open reading frames which take charge of biosynthesis, regulation, and resistance (Figure 6A). A possible biosynthetic pathway for loonamycin was detected based on this bioinformatic analysis (Figure 6B). In particular, compound **27** showed potent cytotoxic activities toward eight cancer cell lines, including Sum1315 (breast cancer), SH-SY5Y (neuroblastoma), HCT116 (colorectal cancer), HT29 (colorectal cancer), HCC78 (lung cancer), HeLa (cervical cancer), SW620 (colorectal cancer), and SW872 (liposarcoma), with IC_50_ values of 41−283 nM [61].

The broth culture crude extracts of actinobacterium *Nocardiopsis* sp. SCA30, derived from marine sediments collected from Havelock Island, Andaman, and the Nicobar Islands, India (11.96° N, 93.00° E), displayed cytotoxic activities against a series of cell lines, including HT 29, HCT 15, MDA-MB 468, and MCF 7 at concentrations ranging from 62.5 to 1000 µg/mL. The strain extracts also showed antibacterial activities against MRSA ATCC NR-46171 and NR-46071 with MIC values of 7.81 and 15.62 µg/mL, respectively. Compound 1-acetyl-4-4(hydroxyphenyl)piperazine (**30**) (Figure 5) was isolated from the crude extracts of *Nocardiopsis* sp. SCA30 through LC-MS analysis and NMR chemical structural identification approved to be an antibacterial and cytotoxic compound [22].

Chemical investigation of actinomycete *N**. dassonvillei* SCSIO 40065 derived from the marine sponge *Petrosia* sp., which was collected on the seabed near Yongxing Island in the South China Sea at a depth of 20 m, led to the isolation of two polycyclic thioalkaloides, dassonmycins A (**31**) and B (**32**) (Figure 5). The new isolated compounds **31** and **32** contained the skeleton of a 6/6/6/6-fused tetracyclic ring featuring a naphthoquinone [2,3-*e*] piperazine-[1,2-*c*] thiomorpholine. Both compounds exhibited antibacterial activities against *Micrococcus luteus* SCSIO ML01, *B. subtilis* 1064, MRSA shhsA1, and *S. aureus* ATCC 29213 with MICs of 8−64 μg/mL. Compound **31** was found to display weak inhibited growth of *Vibrio alginolyticus* ATCC 13214 and *Enterococcus faecalis* ATCC 29212, with MIC values of 32 μg/mL. Compounds **31** and **32** exhibited moderate cytotoxicity against four human cancer cell lines—HepG-2, SF-268, MCF-7, and A549—with IC_50_ values of 12−34 μM [62]. Biosynthetically, **31** and **32** are proposed to be biosynthesized by a non-ribosomal peptide synthetase (NRPS) route combined with a chorismate pathway (Figure 7) [62].

A marine sediment-derived actinobacterium *N. dassonvillei* JS106 showed potent antiquorum sensing activities against *S. aureus* and *Pseudomonas aeruginosa* [63]. The marine sediment sample was gathered from Lianyungang, China. Secondary metabolites research of the strain JS106 led to the isolation of one new compound, 2-hydroxyacetate-3-hydroxyacetamido-phenoxazine (HHP, **33**), and one known analog questiomycin A (**34**) (Figure 5). Both of these two compounds (**33** and **34**) exhibited antibiofilm activity against *Chromobacterium violaceum* 12472 with IC_50_ values of 23.59 and 6.82 μg/mL, respectively [63].

## 4. Peptides

The secondary metabolites research of the broth culture of *Nocardiopsis* sp. UR67A, derived from the marine sponge *Callyspongia* sp. collected from the Red Sea in 2018, resulted in the purification of one new cyclic hexapeptide, nocardiotide A (**35**) (Figure 8) [64]. Compound **35** was synthesized by combining solid-and solution-phase synthesis methods in 2021 [65]. Compound **35** exhibited obvious cytotoxic activities against HeLa, CT26 (murine colon carcinoma), and MM.1S (human multiple myeloma) cell lines with IC_50_ values ranging from 8 to 12 µM [64].

One new diketopiperazine, 1-demethylnocazine A (**36**) (Figure 8), was separated from the broth culture of the actinomycete *Nocardiopsis* sp. TRM20105, which was derived from a local cotton field in Tarim Basin, through the approach of antifungal activity tracking purification against *Candida albicans*. Compound **36** displayed weak antifungal activity against *C. albicans*, with an MIC value of 3.16 mM [66].

One new cyclic tetrapeptide, androsamide (**37**), was isolated from the actinobacterium *Nocardiopsis* sp. CNT-189 was gathered from the surf zone sediment of the Bahamas shore (Figure 8) [67]. Androsamide (**37**) displayed moderate cytotoxicity against Caco2 (human colorectal adenocarcinoma), AGS (human gastric adenocarcinoma), and HCT116 (human colorectal carcinoma) cell lines with the IC_50_ values of 13 μM, 18 μM and 21 μM, respectively. Compound **37** exhibited significant inhibition to the motility of the Caco2 cell line caused by epithelial−mesenchymal transition (EMT) [67]. The expression of cell motility-related genes in Caco2 cells was also examined through Human Cell Motility RT2 Profiler PCR Array. Ras-related C3 botulinum toxin substrate 2 (RAC2) and calpain 1 (CAPN1) have been proved to be the human cell motility-related genes [69,70,71]. The RT2 Profiler PCR Array results showed that **37** with the significant inhibition of the expression of RAC2 in nontoxic concentration 0.65 μM (1/20 of IC_50_), and decreased the expression of CAPN1 by 1.3 μM (1/10 of IC_50_) in Caco2 cells [67].

Secondary metabolites of *Nocardiopsis* sp. HT88, which was separated from the fresh stems of *Mallotus nudiflorus* L. (plant), was comprehensively studied for its crude extracts displayed significant antibacterial activities. Eight proline (or hydroxyproline, Hyp)-containing cyclic dipeptides: cyclo(L-Pro-L-Leu) (**38**), cyclo(Pro-Leu) (**39**), cyclo(L-*trans*-Hyp-L-Leu) (**40**), cyclo(D-*trans*-Hyp-D-Leu) (**41**), cyclo(D-Pro-L-Phe) (**42**), cyclo(L-Pro-L-Phe) (**43**), and cyclo(D-*cis*-Hyp-L-Phe) (**44**), and cyclo(L-*trans*-Hyp-L-Phe) (**45**) (Figure 8) were isolated from the broth culture of HT88 guided by antibacterial bioassay. Unfortunately, none of the purified DKPs (**38**–**45**) exhibited antibacterial activity against measured strains at 100 μg/disc [68].

Some of the peptides are hydrolyzed fragments of a precursor protein, called ribosomal posttranslational peptides (RiPPs) [72], and others are biosynthesized by non-ribosomal peptide synthetase (NRPS) assembly lines (Figure 9) [73,74]. 

There is no research on the biosynthesis of compounds **35** and **37**. The cyclodipeptides’ (CDP) core structures are biosynthesized through NRPS or tRNA-dependent CDP synthase (CDPS) (Figure 10) pathways [75,76].

Compound **36** is belonging to the family of nocazine and is proposed to be synthesized through a CDPS-dependent pathway (Figure 11) [76]. Compounds **38**–**45** are modified CDP and are also deduced to be synthesized by the CDPS assembly line.

## 5. Terphenyls

Chemical investigation of the actinobacterium strain *Nocardiopsis* sp. OUCMDZ-4936, purified from a sample gathered from Dongzhaigang Mangrove Reserve, Hainan Province, China, led to the isolation of three new *p*-terphenyl derivatives, nocarterphenyls A−C (**46**−**48**), together with three known analogs (**49**−**51**) (Figure 12) [77]. Compound **46** exhibited significant cytotoxicity against the HL60 and HCC1954 cancer cell lines, with IC_50_ values of 0.38 ± 0.01 and 0.10 ± 0.01 μM, respectively. Compounds **49** and **51** showed potent cytotoxic activities against K562 and A549 cancer cell lines with IC_50_ values ranging from 0.10 to 0.77 μM. Compound **51** also exhibited cytotoxicity against MV4-11 and MDA-MB-468 cancer cell lines, with IC_50_ values of 0.77 ± 0.02 and 0.67 ± 0.01 μM, respectively [77].

Five new *p*-terphenyl derivatives—called nocarterphenyls D−H (**52**−**56**) (Figure 12)—were gained and identified from the broth culture of the actinobacterium *Nocardiopsis* sp. HDN154086, separated from a marine sediment sample gathered from the South China Sea. It is the first time to find a 2,2′-bithiazole scaffold in natural products. *p*-Terphenylquinones **53** and **54** contain the unusual substitutions of thioether-linked fatty acid methyl ester. Compounds **52** and **53** demonstrated potent activities against seven kinds of bacteria—*E. coli*, *Proteus* sp., *M. phlei*, *V. parahemolyticus*, *B. subtilis*, *B. cereus*, and MRSA—with MIC values of 1.5−12 μM. Notably, compound **53** displayed significant antibacterial activity against MRSA, exceeding the positive control ciprofloxacin [78].

## 6. Others

Three known compounds, tryptophan (**57**), kynurenic acid (**58**), and 4-amino-3-methoxy benzoic acid (**59**) (Figure 13), were isolated from the actinobacterium *Nocardiopsis* sp. UR67, which was derived from the marine sponge *Callyspongia* sp. gained at a depth of 10 m from the Red Sea [64].

One new natural product phenolic acid derivative, 4-amino-6-methylsalicylic acid (**60**), along with two known compounds, 5-methylresorcinol (**61**) and linoleic acid (**62**), were isolated from the *Nocardiopsis* sp. AS23C strain, which was purified from the marine alga *Sargassum arnaudianum* derived from the Red Sea at the Hurghada coast, Egypt (Figure 13). The crude extracts of the strain AS23C displayed antibacterial activities against Gram-positive *B. subtilis* ATCC6051, *S. viridochromogenes* Tü 57, and *S. aureus*. Neither of the compounds **60** and **61** displayed cytotoxicity against KB-3-1 (human cervix carcinoma) cell line [79].

A rare actinomycete *Nocardiopsis* sp. SCA21, deriving from a marine sediment sample gathered from Havelock Island, Andaman, and the Nicobar Islands, India, exhibited the potential ability to produce bioactive secondary metabolites. Chemical investigation of the broth culture of the strain resulted in the purification of two known bioactive compounds—a bromophenol derivative, 4-bromophenol (**63**), and a phthalate ester, Bis (2-ethylhexyl) phthalate (**64**) (Figure 13). Compounds **63** and **64** displayed notable enzyme-inhibitory activities against *α*-glucosidase. However, there was no inhibited activity against *α*-amylase of compound **64**. Both compounds **63** and **64** demonstrated potent free radical scavenging activities against DPPH and ABTS radicals. Additionally, **63** and **64** also revealed broad-spectrum inhibitory activities against MRSA ATCC-46071, MRSA ATCC NR-46171, *S. aureus* ATCC 12600, *B. subtilis* ATCC 6633, and *Klebsiella pneumonia* ATCC 13883 [80].

A novel sterol (**65**) with an unidentified substitution was isolated from marine actinomycete *N. alba* MCCB 110. Compound **65** was found to display antibacterial activity against the aquaculture pathogen *V. harveyi*, and did not exhibit toxicities to the VERO cell line and shrimp hemocytes, up to 1000 ppm [81].

## 7. Conclusions

From March 2018 to 2021, a total of 63 natural products have been isolated from the genus *Nocardiopsis*, and 67% of them are first-discovered compounds. These findings sufficiently demonstrate that actinomycetes *Nocardiopsis* have great potential to produce compounds with novel structures. The structures of the isolated compounds with diverse skeletons are mainly concentrated on the classes of polyketides, peptides, terphenyls, and alkaloids (Table 1 and Figure 14).

The sources of the genus *Nocardiopsis* are distributed throughout diverse ecological systems, including desert, marine, mangrove, and saltpan areas (Figure 15). More than two out of three natural products were isolated from marine-derived *Nocardiopsis*, demonstrating that the ocean is a vast treasure house with abundant microbial sources to produce various novel natural products. From March 2018 to 2021, the percentage of the marine-derived compounds increased by 10%, compared with the period 2015−February 2018 [5], which further demonstrates that more and more attention has been concentrated on the development of the marine resources. Among the marine-derived natural products, the sources of *Nocardiopsis* strains are mainly collected from marine sediment and sponges (Figure 15).

The genus *Nocardiopsis* has the potential ability to produce a great diversity of bioactive secondary metabolites, including antibacterial, cytotoxic, cytoprotective, enzyme inhibitory, free radical scavenging, antiquorum sensing, colorectal cancer motility inhibitory, and antifungal compounds (Figure 16). Almost 50% of the *Nocardiopsis*-derived natural products have been discovered to exhibit various bioactivities. The activities of the bioactive compounds are mainly focused on the categories of antibacterial and cytotoxic activities (Figure 16), which serves as a reminder that there is huge potential for new antibiotics and anticancer compounds to be developed from *Nocardiopsis.*

There were 13 antibacterial natural products isolated from *Nocardiopsis* in the period March 2018 to 2021 (Table 2). Among them, compounds **2**, **30**, and **53** exhibit strong antibacterial activity against MRSA, and compounds **20** and **21** display significant antibacterial activity against MRCNS (Table 2). These five compounds might be developed into new antibiotics to respond to the challenge of increasing antibacterial resistance. Compound **52** shows significant antibacterial activities against a series of strains, which has the potential to be exploited into novel antibiotics with a broad spectrum of antibacterial activity (Table 2). Compound **65** with antibacterial activity displays no toxic against VERO cell line and shrimp hemocytes up to 1000 ppm, could be developed as an environmentally friendly antibiotic (Table 2). All of the isolated antibacterial secondary metabolites provide the structural basis for novel antibiotic research.

There were 13 cytotoxic natural products isolated from *Nocardiopsis* in the period March 2018 to 2021 (Table 3). Compound **27** shows extremely significant and broad-spectrum cytotoxicity against a series of cell lines with the IC_50_ values at the nM level, and compounds **46** and **49**−**51** display potent and broad-spectrum cytotoxicity (Table 3). All four compounds have the potential to be developed as new anticancer drugs. Compounds **8**, **31**, **32**, **35,** and **37** with moderate cytotoxicity provide structural inspiration for the research and development of new anticancer drugs (Table 3).

Eight secondary metabolites isolated from *Nocardiopsis* have various bioactivities, including cytoprotective, antiquorum sensing, colorectal cancer motility and enzyme inhibitory, and free radical scavenging activities (Table 3).

Although the *Nocardiopsis*-derived natural products exhibit excellent diverse bioactivities, further research on their bioactive mechanisms is deficient. Only three compounds, **13**, **16**, and 37**,** have been studied for their active mechanisms between March 2018 and 2021. To the best of our knowledge, all the activity tests were evaluated in vitro, and bioactive evaluations in vivo are also lacking between March 2018 and 2021. In terms of the current situation, it is still a long way from the drugs being approved for clinical use.

In this review, we summarized the secondary metabolites isolated from *Nocardiopsis* in the period between March 2018 and 2021. The literature survey comprehensively indicates that actinomycetes *Nocardiopsis* have great potential as producers to generate abundant and diverse novel bioactive secondary metabolites. Some potent antibacterial and cytotoxic compounds isolated from *Nocardiopsis* have the potential to be developed into new drugs. Additionally, all the natural products isolated from *Nocardiopsis* provide a structural foundation for drug design.

## Figures and Tables

**Figure 1 marinedrugs-20-00374-f001:**
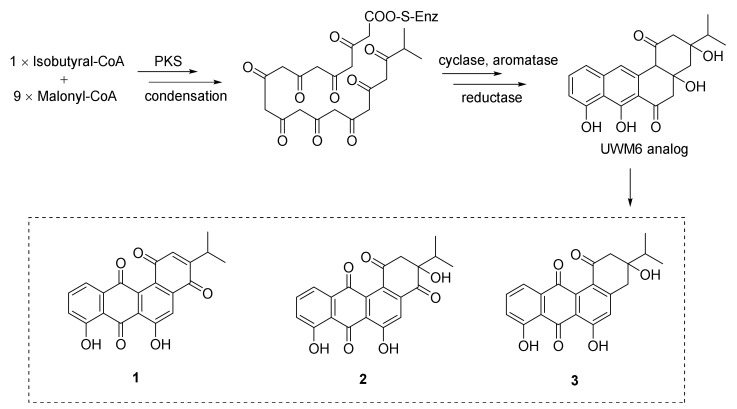
Structural formulas and proposed biosynthesis mechanism for compounds **1**–**3** [48].

**Figure 2 marinedrugs-20-00374-f002:**
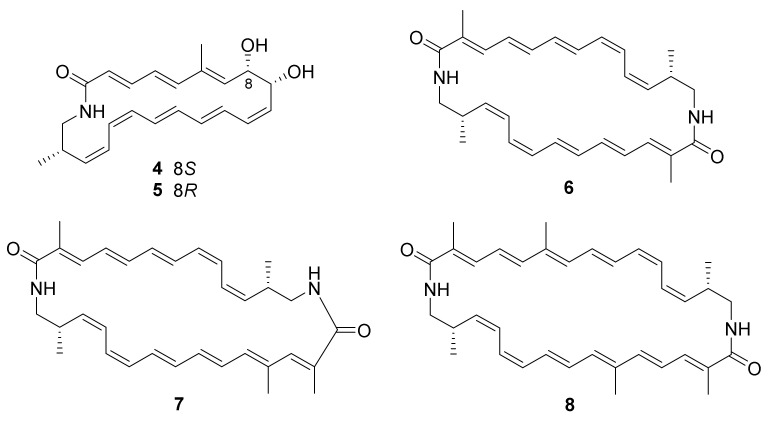
Structural formulas of compounds **4**–**8** [51]. (Reprinted with permission from Ref. [51], Copyright 2019, Journal of Natural Products, published by American Chemical Society).

**Figure 3 marinedrugs-20-00374-f003:**
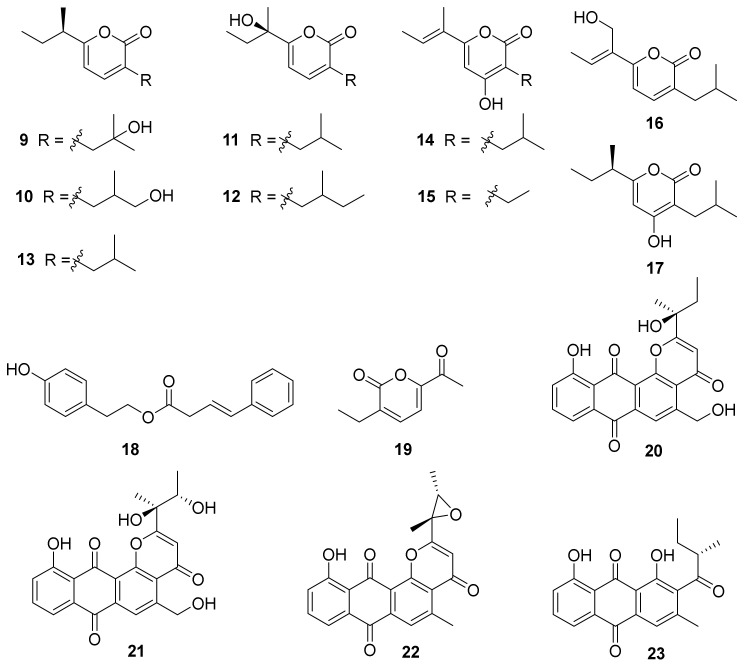
Structural formulas of compounds **9**–**23** [52,55,56].

**Figure 4 marinedrugs-20-00374-f004:**
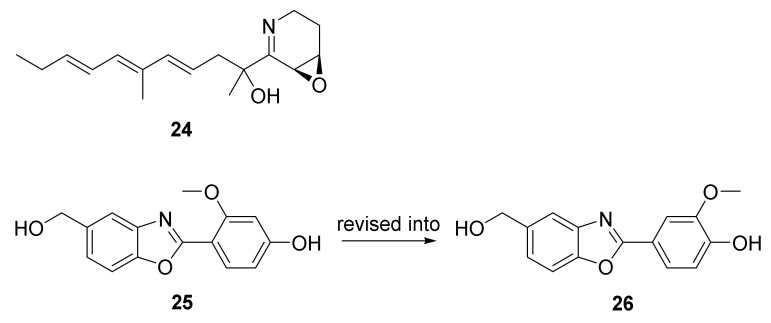
Structural formulas of compounds **24**–**26** [58,59].

**Figure 5 marinedrugs-20-00374-f005:**
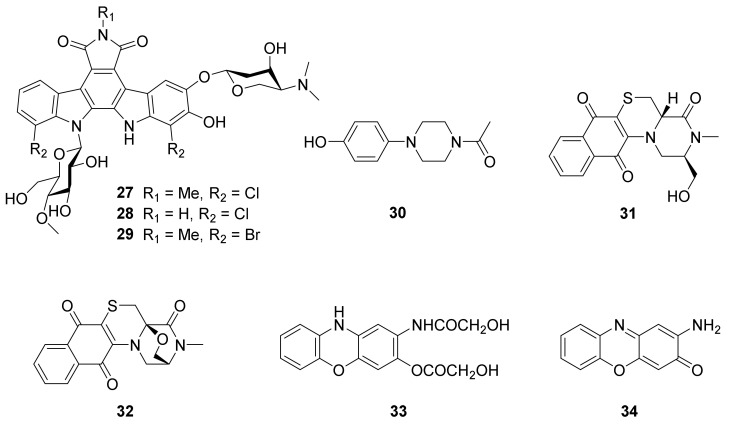
Structural formulas of compounds **27**–**34** [22,61,62,63].

**Figure 6 marinedrugs-20-00374-f006:**
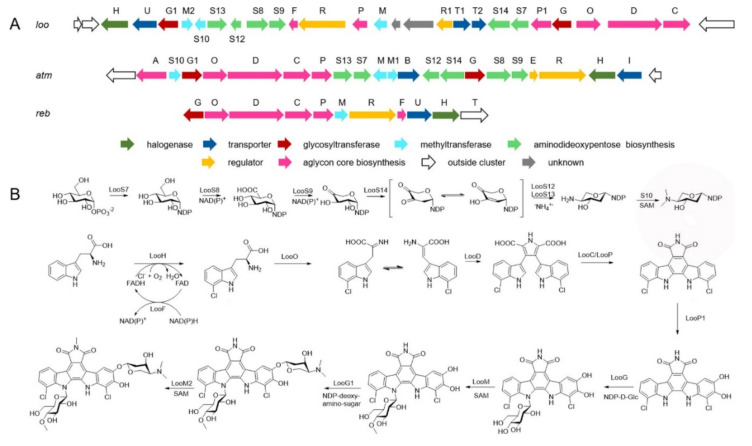
Biosynthesis of loonamycins: (**A**) *loo*, *atm*, and *reb* gene clusters. (**B**) Detected biosynthetic route for compounds **27** and **28** [61]. (Reprinted with permission from Ref. [61], Copyright 2020, Organic Letters, published by American Chemical Society).

**Figure 7 marinedrugs-20-00374-f007:**
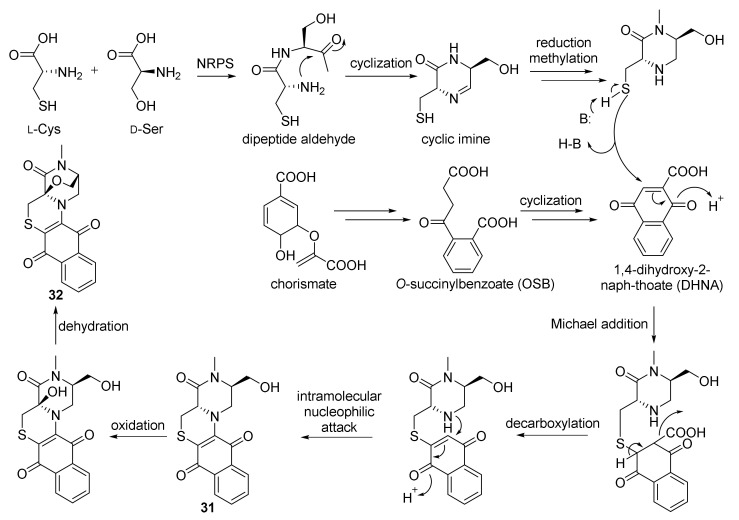
Proposed biosynthesis of compounds **31** and **32** [62]. (Reprinted with permission from Ref. [62], Copyright 2021, Organic Letters, published by American Chemical Society).

**Figure 8 marinedrugs-20-00374-f008:**
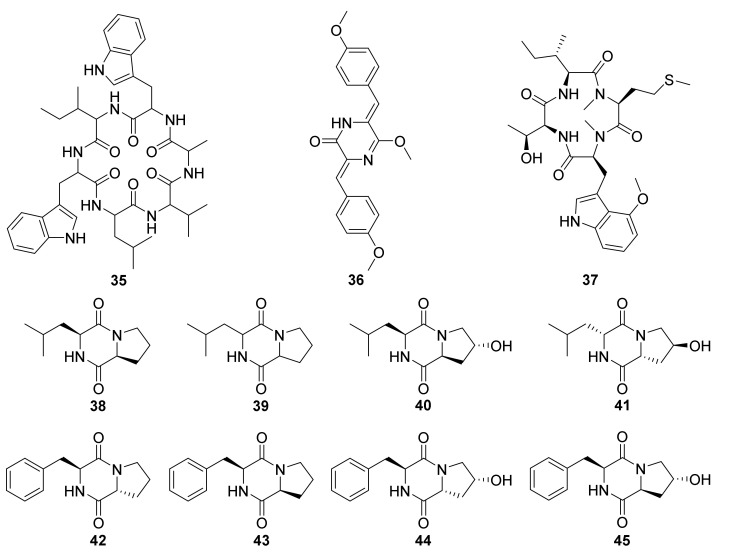
Structural formulas of compounds **35**–**45** [64,66,67,68].

**Figure 9 marinedrugs-20-00374-f009:**
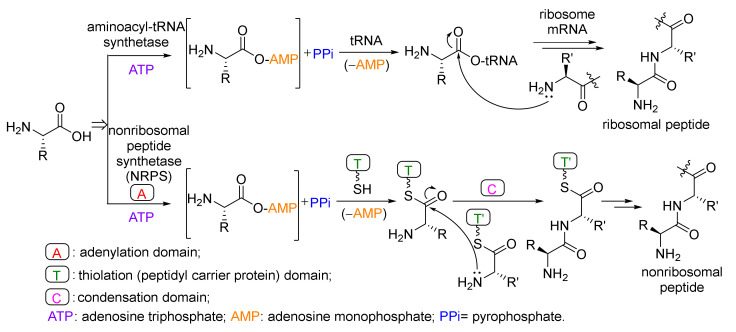
Biosynthesis of ribosomal and non-ribosomal peptides [73].

**Figure 10 marinedrugs-20-00374-f010:**
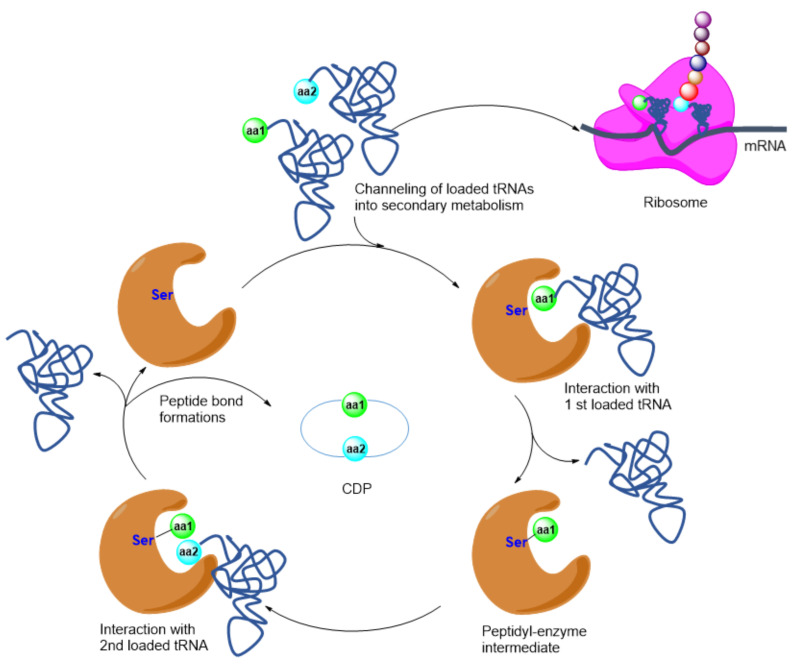
Biosynthesis of CDP through CDPS pathway [76].

**Figure 11 marinedrugs-20-00374-f011:**
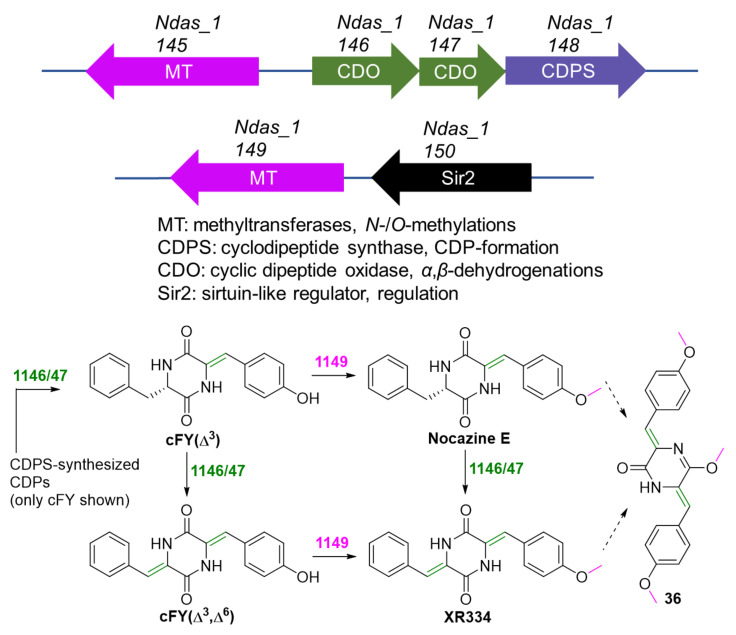
Proposed biosynthesis of compound **36** [76].

**Figure 12 marinedrugs-20-00374-f012:**
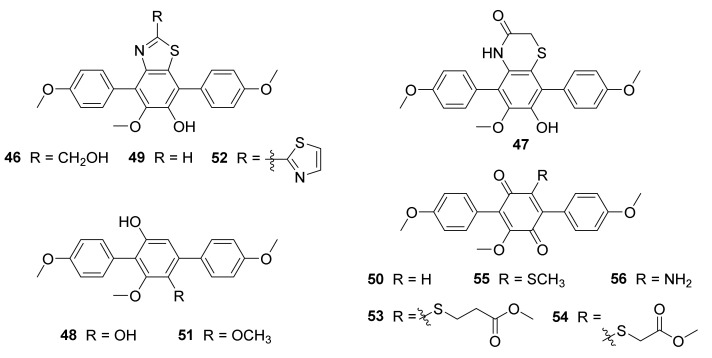
Structural formulas of compounds **46**–**56** [77,78].

**Figure 13 marinedrugs-20-00374-f013:**
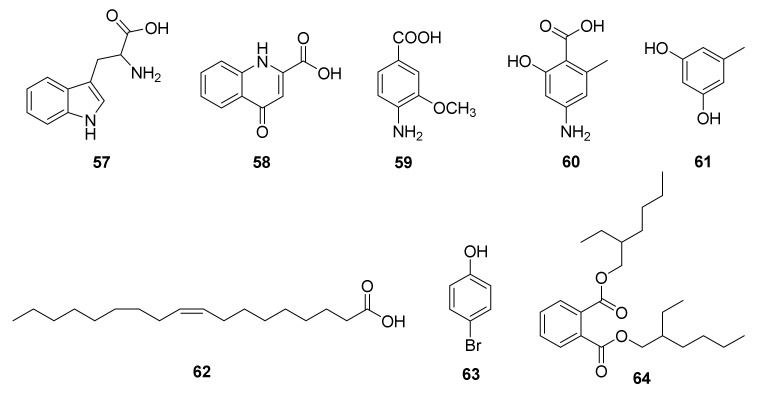
Structural formulas of compounds **57**–**64** [64,79,80].

**Figure 14 marinedrugs-20-00374-f014:**
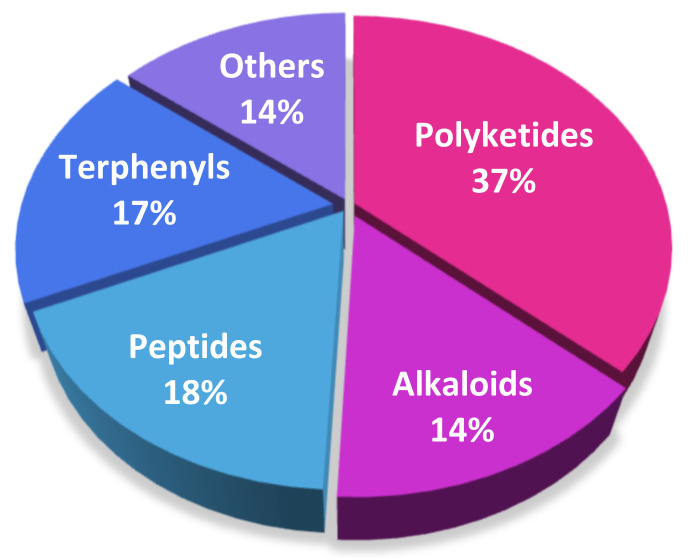
Structural types of compounds isolated from *Nocardiopsis* from March 2018 to 2021.

**Figure 15 marinedrugs-20-00374-f015:**
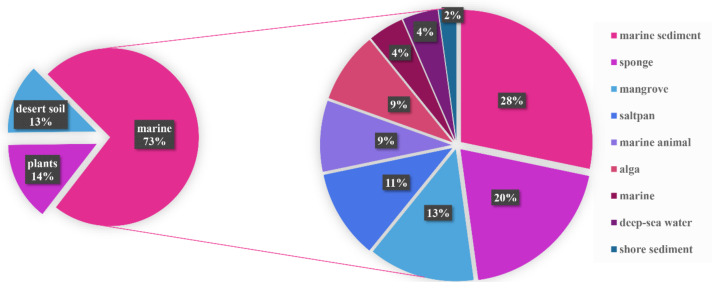
Sources of natural products from *Nocardiopsis* from March 2018 to 2021.

**Figure 16 marinedrugs-20-00374-f016:**
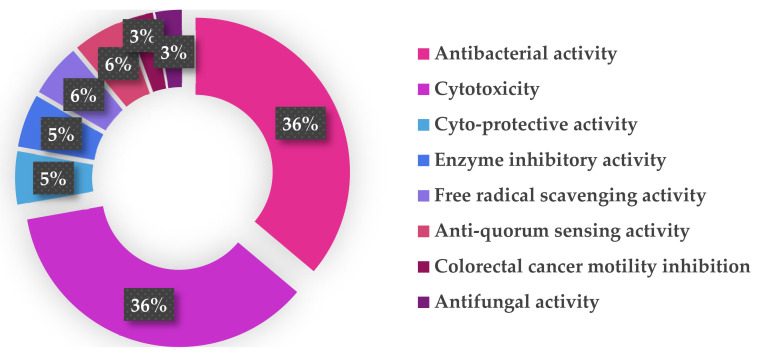
Bioactivities of natural products from *Nocardiopsis* discovered from March 2018 to 2021.

**Table 1 marinedrugs-20-00374-t001:** Compounds isolated from *Nocardiopsis* during March 2018–2021.

Types	Comps.	Sources	Distribution	Bioactivities	Years	Refs
Polyketides	**1–3**	Sponge *Theonella* sp. derived *Nocardiopsis* sp. HB-J378 (GenBank No. MH779065)		Antibacterial activity	2018	[48]
	**4, 5**	Saltpan-derived *Nocardiopsis* sp. CG3 (GenBank No. MG972881)	Kenadsa, Algeria		2019	[51]
	**6−8**	Saltpan-derived *Nocardiopsis* sp. CG3 (GenBank No. MG972881)	Kenadsa, Algeria	Cytotoxicity	2019	[51]
	**9−12, 14, 15, 17**	Desert soil-derived *Nocardiopsis* spp. HDN154-146 (Genbank no. KY794927) and HDN154-168 (Genbank No. MF952649)	Xinjiang, China		2019	[52]
	**13, 16**	Desert soil-derived *Nocardiopsis* spp. HDN154-146 (Genbank no. KY794927) and HDN154-168 (Genbank No. MF952649)	Xinjiang, China	Cytoprotective activity	2019	[52]
	**18, 19**	Deep-sea water-derived *N. dassonvillei* subsp. *albirubida* HDN 17-237 (Genbank No. MN416280)	Mariana Trench		2020	[55]
	**20, 21**	Marine animal-derived *N. aegyptia* HDN19-252 (GenBank No. MN822699)	Antarctic	Antibacterial activity	2021	[56]
	**22, 23**	Marine animal derived strain *N. aegyptia* HDN19-252 (GenBank No. MN822699)	Antarctic		2021	[56]
Alkaloids	**24**	Marine-derived *Nocardiopsis* sp. CNY-503			2018	[58]
	**27**	Marine sediment-derived *N. flavescens* NA01583 (GenBank No. MT371575)	Hainan China	Cytotoxicity	2020	[61]
	**28, 29**	Marine sediment-derived *N. flavescens* NA01583 (GenBank No. MT371575)	Hainan China		2020	[61]
	**30**	Marine sediment-derived *Nocardiopsis* sp. SCA30 (GenBank No. MT573349)	Havelock Island	Antibacterial and anticancer activities	2021	[22]
	**31, 32**	Sponge *Petrosia* sp.-derived *N. dassonvillei* SCSIO 40065 (GenBank No. MW492395)	South China Sea	Antibacterial and cytotoxic activities	2021	[62]
	**33, 34**	Marine sediment-derived *N. dassonvillei* JS106 (GenBank No. MN416229)	Lianyungang, China,	Antiquorum sensing activity	2021	[63]
Peptides	**35**	Sponge *Callyspongia* sp.-derived *Nocardiopsis* sp. UR67	Red Sea	Cytotoxicity	2018	[64]
	**36**	Cotton field-derived *Nocardiopsis* sp. TRM20105	Tarim Basin	Antifungal activity	2019	[66]
	**37**	Shore sediment-derived *Nocardiopsis* sp. CNT-189 (GenBank No. KY111725.1)	Bahamas	Cytotoxicity and colorectal cancer motility inhibitor	2020	[67]
	**38−45**	Stems of *Mallotus nudiflorus* L-derived *Nocardiopsis* sp. HT88 (Genbank No. MH817156)			2020	[68]
Terphenyls	**46, 49−51**	Mangrove-derived *Nocardiopsis* sp. OUCMDZ-4936 (Genbank No. MK129184)	Hainan China	Cytotoxicity	2019	[77]
	**47, 48**	Mangrove-derived *Nocardiopsis* sp. OUCMDZ-4936 (Genbank No. MK129184)	Hainan China		2019	[77]
	**52, 53**	Marine sediment-derived *Nocardiopsis* sp. HDN154086 (GenBank No. MK129184)	South China Sea	Antibacterial activity	2021	[78]
	**54−56**	Marine sediment-derived *Nocardiopsis* sp. HDN154086 (GenBank No. MK129184)	South China Sea		2021	[78]
Others	**57−59**	Sponge *Callyspongia* sp.-derived *Nocardiopsis* sp. UR67	Red Sea		2018	[64]
	**60−62**	Alga *Sargassum arnaudianum*-derived *Nocardiopsis* sp. AS23C (GenBank No. MH144210)	Red Sea		2019	[79]
	**63, 64**	Marine sediment-derived *Nocardiopsis* sp. SCA21 (GenBank No. MH105056)	Havelock Island	Enzyme inhibitory, antibacterial, and free radical scavenging activities	2019	[80]
	**65**	Marine-derived *N. alba* MCCB110 (GenBank No. EU008081)		Antibacterial activity	2021	[81]

**Table 2 marinedrugs-20-00374-t002:** Antibacterial activities of compounds isolated from *Nocardiopsis* from March 2018 to 2021.

Strains	Comps.	Values (MIC)	Pros	Cons
MRSA	**1, 3**	12.5 μg/mL	Specific inhibition of MRSA	Moderate activity [48]
	**2**	3.12 μg/mL	Strong activity; Specific inhibition of MRSA	[48]
MRCNS	**20/21** (μM)	6.2/6.2	Broad-spectrum antibacterial activity; Strong activity against MRCNS compared with positive control	Moderate activity against *B. subtilis* and *Proteus* sp. compared with positive control [56]
*B. subtilis*		6.2/6.2		
*Proteus* sp.		12.5/6.2		
*B. cereus*		6.2/6.2		
*E. coli*		6.2/6.2		
*M. Phlei*		6.2/3.1		
MRSA ATCC NR-46071	**30** (μg/mL)	15.6	Strong activity	[22]
MRSA ATCC NR-46171		7.8		
*B. subtilis* 1064	**31/32** (μg/mL)	8/16	Broad-spectrum antibacterial activity	Weak activity [62]
*M. Luteus* SCSIO ML01		16/32		
*S. aureus* ATCC 29213		64/64		
MRSA shhsA1		32/32		
*E. faecalis* ATCC 29212		32/-		
*V. alginolyticus* 13214		32/64		
*Proteus sp.*	**52/53** (μM)	3.1/12	**52** showed strong and broad-spectrum activity, and **53** displayed strong activity of MRSA	**53** displayed moderate activity against *Proteus* sp. and *B. subtilis* [78]
*B. cereus*		1.5/-		
*M. phlei*		6.2/-		
*B. subtilis*		3.1/12		
MRSA		-/6.2		
*V. parahemolyticus*		6.2/-		
*E. coli*		3.1/-		
*K. pneumoniae* ATCC 13883	**63/64** (μg/mL)	125/250	Broad-spectrum antibacterial activity	Weak activity [80]
*Listeria cytogens* ATCC 13932		62.5/-		
*S. aureus* ATCC 12600		62.5/125		
*B. subtilis* ATCC 6633		7.81/7.81		
MRSA ATCC NR-46171		15.62/7.81		
MRSA ATCC NR-46071		125/15.62		
*V. harveyi* MCCB 111	**65**	20 mm (zone of inhibition)	Not toxic against VERO cell line and shrimp hemocytes up to 1000 ppm	[81]

**Table 3 marinedrugs-20-00374-t003:** Cytotoxicity and other activities of compounds isolated from *Nocardiopsis* from March 2018 to 2021.

Bioactivities	Cells/Stains/Enzyme	Comps.	Values	Pros	Cons
Cytotoxicity (IC_50_)	Hela cells KB3.1	**6/7/8** (μM)	6.8/5.4/2.4	Compound **8** with broad-spectrum cytotoxicity	Moderate activity [51]
	PC-3		6.3/5.0/2.1		
	A549		-/-/6.5		
	SKOV-3		-/10.0/5.5		
	SH-SY5Y	**27** (nM)	283.6	Extremely potent and broad-spectrum cytotoxicity	[61]
	Sum1315		121.3		
	HT29		81.3		
	SW620		90.5		
	HCT116		31.4		
	HeLa		100.1		
	SW872		92.3		
	HCC78		41.5		
		**30**		Broad-spectrum cytotoxicity	IC_50_ untested [22]
	SF-268	**31/32** (μM)	17.0/11.9	Broad-spectrum cytotoxicity	Moderate activity [62]
	MCF-7		25.7/20.7		
	HepG2		31.2/12.0		
	A549		34.4/13.5		
	MM. 1S	**35** (μM)	8	Broad-spectrum cytotoxicity	Moderate activity [64]
	HeLa		11		
	CT26		12		
	AGS	**37** (μM)	13	Broad-spectrum cytotoxicity	Moderate activity [67]
	Caco2		18		
	HCT116		21		
	L-02	**49** (μM)	17	Strong and broad-spectrum cytotoxicity	[77]
	A549		5.1		
	K562		0.77		
	MCF-7		6.0		
	P6C		9.4		
	N87	**46/50/51** (μM)	-/1.0/-		
	A673		-/0.76/8.9		
	MV4-11		4.0/0.16/0.77		
	K562		9.0/4.8/8.9		
	A549		7.8/0.48/9.7		
	BT474		6.0/3.6/-		
	H1229		-/0.72/-		
	HUCCT1		-/0.20/-		
	B16F10		-/0.76/-		
	MDA-MB-468		2.8/1.1/0.67		
	H1975		-/3.1/4.4		
	HL60		0.38/0.17/5.0		
	A431		4.6/-/-		
	U251		-/4.7/-		
	HCC1954		0.10/0.48/2.0		
	MCF-7		18/-/17		
	MKN-45		-/0.49/12		
	DU-145		-/0.52/1.0		
	SPC-A1		-/2.0/9.8		
	HCT116		-/1.9/-		
	143B		5.5/5.0/7.7		
	H2228		1.7/0.94/5.0		
	MDA-MB-231		-/2.0/2.0		
Cytoprotective activity	HaCaT cells	**13, 16**			[52]
Antiquorum sensing activity (IC_50_)	*C. violaceum* 12472	**33/34** (μg/mL)	23.59/6.82		[63]
Antifungal activity (MIC)	*C. albicans*	**36**	3.16 mM		Weak activity [66]
Colorectal cancer motility inhibition	Caco2	**37**		Strong activity	[67]
Enzyme-inhibitory activity (IC_50_)	α-glucosidase	**63/64**	94.61/202.33	strong activity against α-glucosidase	[80]
	α-amylase		103.23/-		
Free radical scavenging activity		**63/64**		Strong activity	[80]

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
