# Peer review of "Genus Nocardiopsis: A Prolific Producer of Natural Products"

_marinedrugs, 2022, doi:10.3390/md20060374_

Round 1

Reviewer 1 Report

The manuscript entitled “Genus Nocardiopsis: A Prolific Producer of Natural Products’’ covers the information regarding the natural products isolated from Nocardiopsis during March 2018 to 2021.  The information is complete and meet the aim that the authors have targeted. 

The authors consider some changes so as to be published. I believe the publication is possible subject:

  • The words in line number 47, chemical structural diverse does not seem to be correct. Instead, you can change it to structurally diverse or similar sounding words.
  • Please correct some spelling errors, rout in line number 68, to has in line 78, the word deriving in line 151 and 185, broad-spectra in line 255.
  • Please re-write the words possible biosynthesis in line 69.
  • The two sentences starting at line number 75 and 76 should be rearranged for the smooth flow of the information. Please, start with second sentence first and then first sentence because second sentence does not look like the continuity of the first.
  • Please re-write the first half of the sentence starting in line 109.
  • The sentence starting at line number 120 talks about the two compounds tryptophan and kynurenic acid. Why did you keep these compounds in alkaloids headings since they are building blocks for alkaloids not the alkaloids themselves?
  • Please provide the bioactivity information of nocarbezoxazole G in the paragraph starting in line 124 since you provide bioactivities for other compounds.
  • The words in line number 138 careful analysis should be re-written.
  • The caption of the figure number 6 (A) in line number 149 should include other gene clusters as that in reference paper since other gene clusters are also included there.
  • Remove the word all in the line 180 and re-write the sentence as both of these or similar sounding words since only two compounds are there.
  • Please re-write the sentence starting at line number 186 especially the words has been successfully total synthesized.
  • The word in Caco2 in line 200 is not the cancer, it’s the cancer cell lines. So, please write the full name for this cell line form the reference paper.
  • The word have should be changed to was in line 252.
  • The table 1 does not provide the information about the Nocardiopsis Since, the table summarizes your information, please provide the information of the bacterial strain. The sources column could be the probable place for information.
  • Please prove the date in the figure 13 caption since you provide the date for other figures.
  • Please include the words anticancer compounds in line 297 after antibiotics since you mention about cytotoxic activity.

Author Response

Dear professor,

Thanks very much for your kind comments, we have revised our manuscript carefully according to your comments point to point. The changes in our new version of manuscript have been highlighted in red.

  1. The “chemical structural diverse” has been revised into “structurally diverse” in line 48.
  2. The spelling errors of “rout” has been revised into “route” in line 68, “has” has been revised into “have” in line 77, “deriving” has been revised into “derived” in lines 156 and 190, “broad-spectra” has been revised into “broad-spectrum” in line 286.
  3. The “possible biosynthesis” has been changed into “were supposed to be biosynthesized” in lines 69 and 70.
  4. The two sentences starting at line 76 and 77 have been rearranged and re-written for the smooth flow of the information as “The new strain Nocardiopsis CG3 (DSM 106572), collected from the saltpan of Kenadsa (22 km west of Bechar, located in southwest Algeria), was discovered to have the ability to produce new bioactive natural products in a screening program. Chemical investigation of the strain led to the isolation of five new polyene macrolactams, kenalactams A−E (48)”.
  5. The first half of the sentence starting in line 117 has been re-written as “These borrelidins were found to have the ability to relieve the amyloid-β induced toxicity in HT22 cell line in 2021”.
  6. The two compounds tryptophan and kynurenic acid have been moved into “Others” heading.
  7. The bioactivity information of nocarbezoxazole G has been provided in lines 132 and 133.
  8. The “careful analysis” has been revised into “carefully analyzing” in line 144.
  9. The caption of Figure 6A has been changed into “loo, atm and reb gene clusters”.
  10. The word “all” has been investigated by “both of these” in line 185.
  11. The sentence has been changed into “Compound 35 was synthesized by combining solid‐and solution‐phase synthesis methods in 2021” in lines 192 and 193.
  12. The full name of Caco2 cell line has been supplied in lines 205 and 206 according to the reference paper.
  13. “Have” has been changed into “was” in line 283.
  14. The information of the bacterial strains Nocardiopsis have been supplied in sources column of Table 1.
  15. The original figure 13 about the nationality distribution of the authors has been deleted according to another reviewer’s comments.
  16. The words “anticancer compounds” have been added in line 321 after antibiotics.

Reviewer 2 Report

  1. The authors include biosynthetic gene clusters for loonamycins but not for other compounds. Authors should include the gene cluster for peptides and non-ribosomal peptides at least.
  2. Authors should include a separate section discussing the mechanisms of action of the described products.
  3. Authors should include the figures, showing different biological activities, and another figure for various mechanisms of action.
  4. Are there any products of this genus under clinical trial or already approved for clinical use? Please include a table regarding such products.
  5. In table1 the authors need to specify if the checked biological activities are in vitro or in vivo.
  6. Please include a table regarding the pros and cons of these compounds, in the way of further development for clinical settings.
  7. What about the in vitro and in vivo cytotoxicity of these compounds? Please discuss.

Author Response

Dear professor,

Thanks very much for your kind comments, we have revised our manuscript carefully according to your comments point to point. The changes in our new version of manuscript have been highlighted in red.

  1. The gene cluster for peptides and non-ribosomal peptides have been supplied in Figures 9−11.
  2. Only three compounds, 13, 16 and 37, have been studied for their active mechanisms between March 2018 and 2021, and their mechanisms have been supplied in lines 90−98 and 208−215.
  3. The different biological activities have been exhibited in Tables 2 and 3. The further research of the compounds' bioactive mechanisms between March 2018 and 2021 are too few to sum up in a table, and have been supplied in lines 90−98 and 208−215.
  4. Our manuscript summarized the isolated compounds in the period between March 2018 and 2021. The time is too short to develop the potentially medicinal compounds into new drugs, so there are no products under clinical trial or already approved for clinical use, this situation has been supplied in line 352.
  5. All the checked biological activities are evaluated in vitro. This has been supplied in line 350.
  6. The pros and cons of these bioactive compounds have been supplied in tables 2 and 3, and discussed in lines 324−334 and 337−346.
  7. All the cytotoxicity of the compounds are evaluated in vitro. This has been supplied in line 350.

Reviewer 3 Report

The author should try to reduce the text similarity within the manuscript

Author Response

Dear professor,

Thanks very much for your kind comments, we have revised our manuscript carefully according to your comments point to point. The changes in our new version of manuscript have been highlighted in red.

We have revised our manuscript according to the plagiarism report to reduce the text similarity. While the most similar places are located in the References part, and in the names of strains, collecting locations and compounds, which couldn’t be modified.

Reviewer 4 Report

I have read manuscript of the review Shi T. et al. ‘Genus Nocardiopsis: A Prolific Producer of Natural Products’ and find it acceptable for publication in Marine Drugs after some minor corrections. The content of the article fits well to the scope of Marine Drugs, and it is a continuum to the series of articles about marine compounds from the Genus Nocardiopsis. The conclusion part’s tables and figures give a nice overview of the compounds isolated from the Genus Nocardiopsis after March 2018.

The manuscript requires a language check.

Chapter starting from line 283 including Figure 13, the nationality distribution of the authors is not relevant, please remove the chapter (lines 283-290) and the figure 13.

Line 11: please remove excellent

Line 47-48: Please rephrase: various chemical structural diverse and bioactive compounds

Line 68: Please replace rout with route

Lines 88-90: Please clarify the sentence, what do you mean by “cyto-protective activity for the first time to be found”?

Lines 138-140: Please clarify the sentence

Line 244: What is obvious cytotoxicity? Please clarify.

Line 259: no obvious cytotoxicity? Please clarify

References are missing from the Figure 3, 4, 5, 8, 9, 10 captions, please add.

Author Response

Dear professor,

Thanks very much for your kind comments, we have revised our manuscript carefully according to your comments point to point. The changes in our new version of manuscript have been highlighted in red.

  1. We have checked our English language carefully, and revised some wrong expressions.
  2. The original Figure 13 about the nationality distribution of the authors, and the relevant sentences have been removed.
  3. “Excellent” has been removed in line 11.
  4. The sentence “various chemical structural diverse and bioactive compounds” has been changed into “bioactive and structurally diverse compounds” in line 48.
  5. “rout” has been revised into “route” in line 68.
  6. The sentence of “cyto-protective activity for the first time to be found” has been clarified into “This was the first time to report that compounds 13 and 16 showed cyto-protective activity through inducing the expression of phase II detoxifying enzymes”, in lines 90–92.
  7. The sentences in lines 141–145 have been revised into “The indolocarbazole skeleton of 2729, belongs to the family of indolocarbazole alkaloid, is structurally similar to that of rebeccamycin and staurosporine with additional rare modified tryptophan ring. The molecular basis of these modifications were investigated by carefully analyzing loo BGC for further genome mining and combinational biosynthetic research”.
  8. The sentence has been clarified into “Neither of the compounds 60 and 61 displayed cytotoxicity” in lines 274 and 275.
  9. The sentence has been revised into “and didn’t exhibited toxicities” in line 290.
  10. The references of Figures 3, 4, 5, 8, 12, 13 captions have been supplied.

Round 2

Reviewer 2 Report

The authors successfully responded to the reviewer's comments/suggestions.